# Fundamental and Clinical Applications of Materials Based on Cancer-Associated Fibroblasts in Cancers

**DOI:** 10.3390/ijms222111671

**Published:** 2021-10-28

**Authors:** Jingtian Mu, Shengtao Gao, Jin Yang, Fanglong Wu, Hongmei Zhou

**Affiliations:** 1State Key Laboratory of Oral Diseases, National Center of Stomatology, National Clinical Research Center for Oral Diseases, Frontier Innovation Center for Dental Medicine Plus, West China Hospital of Stomatology, Sichuan University, Chengdu 610041, China; jingtianmou@163.com (J.M.); yangjin@scu.edu.cn (J.Y.); 2State Key Laboratory of Oral Diseases, West China College of Stomatology, Sichuan University, Chengdu 610041, China; scugaoshengtao@163.com

**Keywords:** cancer-associated fibroblasts, cancer, biochip, three-dimensional co-cultivation, nanomaterials

## Abstract

Cancer stromal cells play a role in promoting tumor relapse and therapeutic resistance. Therefore, the current treatment paradigms for cancers are usually insufficient to eradicate cancer cells, and anti-cancer therapeutic strategies targeting stromal cells have been developed. Cancer-associated fibroblasts (CAFs) are perpetually activated fibroblasts in the tumor stroma. CAFs are the most abundant and highly heterogeneous stromal cells, and they are critically involved in cancer occurrence and progression. These effects are due to their various roles in the remodeling of the extracellular matrix, maintenance of cancer stemness, modulation of tumor metabolism, and promotion of therapy resistance. Recently, biomaterials and nanomaterials based on CAFs have been increasingly developed to perform gene or protein expression analysis, three-dimensional (3D) co-cultivation, and targeted drug delivery in cancer treatment. In this review, we systematically summarize the current research to fully understand the relevant materials and their functional diversity in CAFs, and we highlight the potential clinical applications of CAFs-oriented biomaterials and nanomaterials in anti-cancer therapy.

## 1. Introduction

Cancer-associated fibroblasts (CAFs), as a critical component of the tumor stroma, are strong promoters of various tumor behaviors, including tumorigenesis, growth, invasion, and/or metastasis, because they produce abundant extracellular matrices (ECMs) and mediate the proliferation, apoptosis, migration, and stemness of tumor cells [1,2]. Clinically, numerous studies have shown that CAFs can reduce the efficacy of a variety of anti-tumor treatments, including chemotherapy, radiotherapy, biotherapy, and/or targeted therapy, subsequently leading to therapeutic resistance or even failure [3]. For instance, CAFs can prevent the penetration of chemotherapeutic drugs by synthesizing ECMs, promote the growth and metabolism of tumor cells by paracrine signaling/exosome secretion, and prevent the eradication of tumors by therapeutic agents [4,5]. Further, CAFs can reduce the sensitivity of tumors to radiotherapy by promoting the epithelial–mesenchymal transition (EMT) of tumor cells and the survival of cancer stem cells (CSCs) [6,7]. Thus, in the past decades, the design, improvement, and application of biological materials targeting CAFs have attracted much attention for their potential to optimize the therapeutic efficacy of anti-cancer treatments.

At present, studies on biological materials targeting CAFs can generally be divided into three aspects depending on their purpose: high-throughput screening and sequencing, three-dimensional (3D) culture technologies, and nanomaterials for therapeutic applications. Recently, by using a variety of microarrays, differential expression/infiltration patterns of CAFs were detected and analyzed at gene, protein, and tissue levels [8,9,10]. Differentially expressed genes and/or proteins may be targeted as therapeutic markers of crosstalk between CAFs and tumor cells, or applied as indicators for evaluating different functional therapeutic parameters, such as efficacy, complications, and survival rate. Moreover, by improving and optimizing cultivation technology or scaffold materials, 3D culture models of in vivo tumors have been used to explore different effects of CAFs on tumor cells [11]. Importantly, several metal nanoparticles and drug-carrying systems have been developed to target and eliminate CAFs or inhibit their functions to improve anti-cancer therapy [12].

Although the application trend of these biological materials targeting CAFs is promising for anti-tumor treatment, huge challenges must be overcome before they can be used in clinical practice. Due to the fact that the spatial structures differ between existing scaffold materials in co-culture models and the natural ECM, the cell composition of these models, typically from cell lines, is relatively uniform [13], which greatly differs from the organization of tumor tissues. Additionally, nanotherapy, when targeting CAFs, has encountered therapeutic resistance and has even caused adverse reactions in bone marrow [14,15], thus largely limiting its clinical applications. In this review, we systematically summarize recent findings related to biological materials targeting CAFs, from basic research to the clinical setting, to better understand their applications. We then offer new perspectives in this field and propose potential treatment strategies.

## 2. Gene Chips and Protein Chips Targeting CAFs

To date, one of the most difficult aspects of targeting CAFs in anti-tumor treatment design has been the difficultly in locating the “Achilles’ heel” among thousands of mutant genes and proteins. Thus, biochips have been widely used to rapidly and accurately identify potential candidates in CAFs for anti-tumor treatments (Figure 1). Typically, a biochip consists of an array of molecular sensors placed on a small surface with a strong substrate to analyze genes, proteins, and/or tissues, enabling the simultaneous execution of many biological or chemical reactions in a relatively short time in a high-throughput process [16]. Different types of biochips, including gene and protein chips, are basically composed of immobilized biomolecules and solid support materials to analyze CAFs [10,16,17,18]. In general, the sample preparation and testing procedures for the biochip analysis of CAFs are similar to those for other cells [19,20]; however, the analysis of CAFs with biochips still exhibits some idiosyncrasies when compared to other types of cells.

### 2.1. Selection of Samples: From Clinical Specimens to Primary Culture

CAFs and normal fibroblasts (NFs) derived from primary culture are the main cellular sources for biochip detection (Figure 1). The majority of CAFs are isolated from fresh tumor specimens. To analyze differences in profiles, it is necessary to use NFs as controls (Table 1). Since the tumor and its surrounding tissues are usually surgically removed together, tumor-adjacent tissue located more than 2–5 cm away from the primary site is the most common source of NFs [21,22,23]. Additionally, NFs can be obtained from the skin of the head and neck, gingiva, buccal mucosa, or foreskin tissue after circumcision [24,25,26]. Further, commercial cell lines of NFs can be used as controls. For instance, NFs purchased from Jingkang (Shanghai, China) were used to study the miRNA differences in CAF-derived exosomes in breast cancer [7]. The obtained expression profiles were very sensitive to the controls; even in the same study, using NFs from different sources led to different biochip results. In support of this finding, Enkelmann et al. found that miR-16 was up-regulated in bladder CAFs when compared with foreskin fibroblasts, but not when compared with fibroblasts from normal urothelial tissue [25]. Intriguingly, sequence analysis further revealed that miR-16 was decreased in foreskin tissue compared with urothelial tissue, which may explain this discrepancy [25]. These results suggest that miR-16 expression in bladder CAFs may not be significantly changed, and the agents targeting miR-16 in bladder CAFs might be off-target or less effective. Similarly, Nakagawa et al. used NFs from the liver and skin as controls in separate analyses, and the results showed that reticulon 1 (RTN1), dickkopf homolog 1 (DKK1), hypothetical protein PP1044 (PP1044), and cyclin-dependent kinase inhibitor 2A (CDKN2A) were up-regulated in CAFs when compared with liver NFs, whereas proteoglycan 1 (PRG1), ankyrin 3 (ANK3), and monocyte chemotactic protein 1 (MCP1) were increased when compared with skin NFs [19]. Notably, vascular cell adhesion molecule 1 (VCAM1) was up-regulated in CAFs compared with NFs from both the liver and skin [19], indicating that VCAM1 was consistently increased in CAFs and might be a candidate target for anti-tumor therapy in metastatic colon cancer. In summary, these findings show that CAFs and NFs have a high cellular heterogeneity, and the expression profiles of NFs are variable depending on their tissue source. Accurately obtaining potential candidate targets in CAFs requires control NFs from at least two sources during high-throughput screening, or the preliminary results from biochips need to be re-validated by other methods before processing.

### 2.2. Before Testing: Primary Culture and Identification of CAFs

Typically, tissues are rinsed with phosphate-buffered saline (PBS) containing penicillin and streptomycin, after which, they are cut into 1–3 cm^3^ pieces and further digested by protease for use in the primary culture. Cellular components and/or conditioned media are used for hybridization with the biochip. Protease digestion is used to isolate CAFs, and the most commonly used enzymes are trypsin and collagenase (type I or IV) [20,23]. Ethylene diamine tetra-acetic acid (EDTA) can bind calcium ions and decrease cell adhesion, so it is widely used in enzyme digestion [24]. Fibroblasts are easy to grow and adherent to plastic. Pure CAFs can be obtained by exploiting differences in the adherence time between CAFs and tumor cells. Specifically, fibroblasts can be obtained by removing the non-adherent tumor cells after 30 min (because the adherence time is <30 min for fibroblasts but >1 h for tumor cells) [17].

To date, no special culture medium has been developed for CAFs. Dulbecco’s modified Eagle’s medium (DMEM) (GIBCO, Grand Island, NY, USA) containing 10% fetal bovine serum (FBS) and antibiotics (complete medium) is typically used to culture CAFs and NFs [18,23]. Ham’s nutrient mixture F12 has been added to this medium to further support fibroblast growth [32]. In addition, Roswell Park Memorial Institute (RPMI) (HyClone, Logan, UT, USA) 1640 medium with 1% FBS has also been used for fibroblast culture [17]. Additionally, Eagle’s minimum essential medium (EMEM) with 15% FBS has been specialized to support fibroblast growth for the isolation and culture of CAFs, and Medium 199 and Ham’s F12 mixed in a 1:1 ratio have the same function [10,33]. To ensure the accuracy of the analysis, it is necessary to identify CAFs before the high-throughput screening and sequencing by biochips. Commonly used methods for CAF identification mainly include morphological observation, immunohistochemistry, and immunofluorescence. CAFs have a negative reaction to epithelial-derived markers and a positive reaction to mesenchymal markers. We previously reported, that compared with NFs, CAFs highly express α-SMA, fibroblast-activated protein (FAP), and/or PDGF receptor α/β (PDGFRα/β) [2].

In most studies, CAFs were passaged three to five times before analysis, while 10 passages were considered acceptable for primary cells [23]. The total RNA of the primary cultured fibroblasts was extracted by Trizol Reagent (Invitrogen) or commercial kits, such as the RNeasy Mini kit (Qiagen, Inc., Valencia, CA, USA) and miRVana Isolation Kit (Ambion, Life Technologies, Milan, Italy) [7,21,28]. For the protein arrays, the conditioned medium was collected after 2000× *g* centrifugation and stored at –80 °C if hybridization with biochips could not be performed immediately [17]. Recently, many studies have tended to use arrays from Affymetrix (Thermo Fisher Scientific Inc., Santa Clara, CA, USA) and Agilent (Agilent Technologies, Santa Clara, CA, USA) to explore mRNA and/or miRNA expression [22,30], and, to date, the cytokine antibody chip from RayBio (RayBio, AAH-BLG-493) has mainly been used to detect the level of protein secreted by CAFs [20,34]. In summary, CAF analysis using biochips depends on cell isolation, culture, and identification. As a result of cellular heterogeneity, all or a combination of α-SMA, FAP, and/or PDGFRα/β should be used for identification. Biochips from different manufacturers, such as Affymetrix, Agilent, and RayBio, need to be chosen according to the type of expression profile being analyzed (Table 1).

### 2.3. After the Array Analysis: Validating Gene or Protein Expression Levels

The differential expression obtained from biochips reflects relative changes and needs to be validated by quantitative methods. For gene chips, quantitative polymerase chain reaction (qPCR) is commonly used, and western blotting (WB) and/or enzyme-linked immunosorbent assay (ELISA) are used in the detection of protein levels (Table 1). In most cases, the qPCR or WB results have corresponded to the gene expression obtained in the biochip analysis. Zhao et al. randomly chose miRNAs for further analysis by RT-qPCR, and the data were highly similar to the biochip results [23]. Similarly, in another study, ELISA and WB indicated that the concentration of plasminogen activator inhibitor-1(PAI-1) was higher in the conditioned medium of CAFs, which was also observed in the cytokine array results [17]. However, Enkelmann et al. provided evidence that, whereas biochip results indicated that miR143 and miR145 were down-regulated in CAFs in bladder cancer, qPCR showed no notable changes in the same miRNAs [25], indicating that foreskin fibroblasts might not be suitable as a control in this study, or that the difference might be caused by a sample size bias. Supporting this notion, Utaijaratrasmi et al. found that the expression of miR-486 was different in two normal skin fibroblasts [24], suggesting that the sample size needs to be expanded when discrepancies appear. Taken together, the expression profiles of CAFs from biochips can strongly vary as a result of different factors, from the sample size to controls; thus, further validation by other methods needs to be performed to confirm the results obtained using biochips.

## 3. Three-Dimensional Co-Cultivation Materials

The tumor microenvironment (TME) is complex in composition and structure, which can affect tumor growth, metastasis, and the cellular phenotypes of tumor and stromal cells [35]. Traditional co-cultivation techniques have been primarily carried out on a two-dimensional (2D) plane, which is quite different from the growth mode of cells in vivo [36], leading to deviations from the actual situation in terms of cell growth, differentiation, and interaction. In recent years, with the development of materials research, the emergence of multiple scaffold materials has made it possible to construct 3D co-culture models for in vitro studies in cell, tissue, and/or organ cultures, and 3D co-cultivation appears to have a promising future in facilitating the translation of basic research to the clinical setting. At present, the types of 3D co-culture models related to CAFs can generally be divided into scaffold-free co-culture systems, scaffold-based co-culture models, and microfluidic platform co-cultivation technology (Figure 2).

### 3.1. Scaffold-Free Culture of CAFs

Due to buoyancy and gravity, suspended cells can aggregate into small spherical cell clumps without the aid of foreign scaffold materials [37]; therefore, this method has been widely used to explore the role of CAFs in tumor growth using hanging-drop and low-adherence plates. For instance, due to the effect of surface tension, liquid can hang onto the lid to form hanging drops, and the cells therein can gather into a spherical shape under the action of gravity [36]. Similarly, by mixing and adding the same number of cancer cells and CAFs from the primary culture into hanging-drop plates, Ma et al. transferred and harvested a cell suspension containing spheroids after 72 h. After imaging by microscopy, the authors found that CAFs could promote the growth of gastric cancer cells and increase the diameter of the spheres [38]. Further, when a well plate is covered with an inert substrate (usually polystyrene), cells are unable to adhere to the plate wall, and the suspended cells aggregate into visible spheroids. Zhou et al. harvested similar spheroids consisting of melanoma cells and CAFs of melanoma for 48 h by inoculating a mixture of cell suspension into 96-well plates with low-cell-adhesion surfaces [39]. Although centrifugation will accelerate the formation of spheroids, it is limited in the hanging drop, which increases the experimental period. Taken together, in contrast to traditional direct or indirect co-culture, the scaffold-free culture allows cell spheres to grow and form a spatial structure, which is more similar to tumors in vivo and can be used to evaluate the promoting effect of CAFs on tumor growth. Notably, when the cell spheroids are suspended in the medium, their motility cannot be evaluated or controlled, which limits the further analysis of the role of CAFs in tumor behaviors, such as invasion, metastasis, etc.

### 3.2. Scaffold-Based Culture in CAFs

The scaffold material with a complex 3D structure provides support to cells seeded therein [40], suggesting that models with more complex scaffold and cell structures are feasible. Recently, three types of materials have generally been applied in studies on CAFs. Firstly, natural scaffold material, represented by Matrigel, which is extracted from animal tumors, has the most similar structure to natural tumors [41], but its composition is relatively fixed. Secondly, rat-tail collagen is a type of natural scaffold material, and, unlike Matrigel, its concentration can be adjusted individually in different studies [42]. Thirdly, artificially synthesized scaffold materials are also used for the establishment of CAFs co-cultivation models. Unlike the two aforementioned natural materials, synthetic materials provide good support with no potential immunogenicity [43]. Typically, CAFs derived from the primary culture of tumor tissues are seeded into scaffold materials. Notably, the ratio of tumor cells to CAFs needs to be maintained at 1:1 to 1:3 during tumor formation, since the addition of excessive CAFs may cause fibrosis [44,45], subsequently leading to the failure of the tumorigenesis experiment. (Table 2).

Since a scaffold material with a porous structure provides space and support for the growth of cells, the growth patterns of cells in a scaffold-based culture differ from those in a 2D co-culture. For example, Phan-Lai et al. co-cultured mouse mammary carcinoma (MMC) cells and mouse CAFs in both 2D plates and chitosan–alginate scaffold, and they observed that MMC cells formed tumor spheroids and had slower growth in the scaffold-based culture than in the 2D co-culture [46]. Intriguingly, scaffold-based culture technology can be combined with a scaffold-free culture to explore the role of CAFs in tumor cell invasion and migration. For instance, Pankova et al. used the hanging-drop method to acquire tumor spheroids, and then embedded them in rat-tail collagen gel pre-inoculated with CAFs, which increased the protrusions of spheroids and enhanced the invasive activity of cancer cells [47]. These findings suggest that the scaffold-based co-culture has multiple advantages, including the rapid formation of spheroids, well-controlled cellular motility, and the 3D structure and spatial organization, in order to study tumorigenesis, invasion, and metastasis.

**Table 2 ijms-22-11671-t002:** 3D co-culture in the cancer-associated fibroblasts in cancers.

Tumor	Source of CAFs	Scaffold	Scaffold Concentration	TCs: CAFs	Effects on Biological Behaviors	Ref.
ACC	Primary culture	BME	N/A	1:1	Increase invasion	[45]
AB	Primary culture	Rat-tail collagen	N/A	1:3	Support proliferation, invasion	[44]
BC	Primary cultureFlow cytometryCell lines	CA Matrigel PCL/Scaffold-free	4% CA 1 mg/mL	2:1 to 1:5	Enhanced growth, survival, invasive, migration, and proinflammatory cytokines	[41,43,46,48,49]
CRC	Primary cultureNFs co-culture with CRC	Matrigel Rat-tail collagen	8–11 mg/mL	1:1	Enhanced metastasis and adhesion. Inhibit vascular-like network formation	[50,51]
ESCA	Primary culture	Collagen-1 gels	N/A	N/A	Promote invasion	[52]
GC	Primary culture	Scaffold-free	N/A	1:1	Enhance growth	[38]
GBC	Primary culture	Matrigel Rat-tail collagen	N/A	1:1	Promote proliferation, invasion, migration, and tube formation	[53]
HNSCC	Primary culture	Hydrogel scaffold	2.55–5.11 mg/mL	1:1	Enhance invasion of CAL33 cells	[54]
OSCC	Primary culture	Scaffold-free	N/A	1:1	Promote stemness	[55]
NSCLC	Primary cultureCell lines	Rat-tail collagen Collagen gel BME/Cultrex matrix	2.0 mg/mL	1:1	Increase proliferation, migration, invasion, chemoresistance, and contraction	[47,56,57,58,59]
MEL	Col1α2-CreER fibroblasts	Scaffold-free	N/A	1:2	Enhanced growth	[11,39]
PC	Primary cultureCell lines	ECM Collagen lattice	1 mg/mL	5:1	Increase growth, invasion, motility migration, and contraction	[60,61]
PDAC	Primary culture	Matrigel	9.2 mg/mL	2:1	Induce therapeutic resistance	[62]

ACC: adenoid cystic carcinoma, AB: ameloblatoma, BC: breast cancer, BME: basement membrane extract, TCs: tumor cells, CA: chitosan-alginate, CAFs: cancer-associated fibroblasts, CRC: colorectal cancer, ECM: extracellular matrix, ESCA: esophageal carcinoma, GBC: gallbladder cancer, HNSCC: head and neck squamous cell carcinoma, N/A: not available, NFs: normal fibroblasts, MEL: melanoma, NSCLC: non-small-cell lung cancer, OSCC: oral squamous cell carcinoma, PCL: polyepsilon-caprolactone.

Furthermore, the scaffold material can not only serve as a supportive function but can also form a more complex layered structure that is similar to the skin [36]. Most tumors, especially cancers from the epithelium, are divided into epithelial and mesenchymal layers. CAFs are mainly located in the mesenchyme and interact with tumor cells. Theoretically, by mixing CAFs with a gel and allowing it to solidify, the surface layer of CAFs can be inoculated with a layer of tumor cells, which results in a hierarchical structure similar to that of epithelial tumors in vivo. To support this notion, Chantravekin et al. seeded epithelial cells on the surface of a gel containing ameloblastoma-associated fibroblasts, and the surface became white because of the multiplication of epithelial cells [44]. Similarly, since cells in the top layer are usually exposed to the air and only receive growth support from the gel containing CAFs below, which simulates the actual TME, Horie et al. exposed the gels to the air by placing them on a mesh in new plates with a growth medium for 5 days. Invasion and nodular epithelial structures were observed in the CAF layer [57], indicating that CAFs enhanced the invasion of tumor cells and might induce the differentiation of lung cancer cells into mucinous cells.

As a type of fibroblasts, CAFs play roles in ECM synthesis and tissue contraction [63], and 3D co-cultivation has made it possible to explore these functions. Fibroblasts can ingest exosomes and be converted into CAFs [64]. However, tumor cells and CAFs in vivo are not simply immersed in liquid and grow in a single layer; they are surrounded by rich ECM, which will affect the delivery of exosomes. For instance, Jung et al. used type I collagen as a scaffold to co-culture breast cancer cells with NFs and found that the radially arranged ECM fibers of tumor spheres enhanced the spread of exosomes to promote the transformation of NFs into CAFs, and that this process was inhibited by blocking the alignment of the ECM [65]. In another study by Chantravekin et al., after 7 days of incubating fabricated gel, the gel had shrunk to approximately half of its original size because of the action of fibroblasts on the collagen [44]. These data suggest that using a 3D co-culture can replicate the shrinkage of tumor tissue mediated by CAFs in vitro.

### 3.3. Microfluidic Platform Co-Cultivation Technology in CAFs

A microfluidic platform is a type of co-cultivation technology and is similar to the scaffold-based culture [66]. Similar to the scaffold-based co-culture, the microfluidic platform provides support for the growth and interaction of tumor cells and CAFs in a 3D space [67]. However, the microfluidic platform has the advantages in the high-throughput culture and simulation of mechanical stress. Etching technology can be used to create multiple cell compartment structures and liquid channels on the surface of a carrier (glass or silicon-based material). These dense channels provide a large space for the growth of cells, thus being beneficial for high-throughput cell culture. For instance, Chen et al. conducted a microfluidic sphere formation platform that generate 1024 uniform cancer spheres within a 2 cm area, where the subsequent co-culture showed that the presence of CAFs can reduce the sensitivity of multiple spheres to cisplatin, and that there was no significant increase in photodynamic therapy (PDT) resistance [68]. Another advantage of the microfluidic platform is that it can simulate the mechanical stress in TME. The liquid channels on the surface of a carrier can be connected with a perfusion system to form an environment with fluid pressure, addressing the insufficiency of the traditional scaffold-based culture [69,70]. Using liquid channels in a microfluidic platform, Yeon et al. imitated the interstitial fluid flow (IFF) in tumors and verified that exosomes from melanoma cells were delivered to the stroma via IFF, and that the number of CAFs differentiated from human umbilical vein endothelial cells increased [11]. In addition to simulating IFF, the microfluidic platform can also be used to simulate the effect of tensile stress on CAFs and tumor cells in TME. Ao et al. used a microfluidic platform to bring mechanical stretching on the normal tissue-associated fibroblasts (NAFs). Similar to CAFs, the stretching NAFs produced ECM with a more organized structure, and guided and increased the cancer cell migration [71].

In summary, these different types of 3D co-culture models have been developed to explore the different roles of CAFs in tumor progression. The scaffold-free culture has made it possible to study tumor growth induced by CAFs in a 3D configuration. Various scaffold materials imitate the complex layered structure and can be used to study tumorigenesis and invasion, which is limited in the scaffold-free culture. In addition to the inherent advantages of the scaffold co-culture, the microfluidic platform effectively simulates the fluid pressure in tumor tissues; therefore, these systems can be used to study the differentiation and function of CAFs under the influence of fluid pressure. Indeed, there are abundant culture models that can be selected to study CAFs, and, thus, it is necessary to select the appropriate model according to the purpose of the study.

## 4. Nanomaterials Targeting CAFs

Accumulating evidence suggests that the response to anti-tumor therapies highly depends on the features of the tumor cells and the TME [72]. As described above, CAFs, as the major cells in the TME, can secrete a variety of soluble cytokines and/or exosomes to support tumor survival, promote metastasis, and enable tumor cells to escape immune attacks [73]. Therapies targeting CAFs might improve the efficacy of anti-tumor treatment and reduce therapeutic resistance [74]. However, the ECM and fluid pressure formed by CAFs create a physical barrier that restricts the entry of drugs [75,76]. Recently, a variety of materials have been successfully prepared to target CAFs for anti-cancer treatment, and have shown therapeutic effects [77,78] (Table 3). These data have shown that nanoparticles or drug delivery systems designed to target the functions of CAFs exhibit promising therapeutic effects. Although different nanosystems considerably vary in their composition and mechanism, they mainly consist of three parts: targeting ligands, drug-carrying systems, and cargo (Figure 3).

### 4.1. Targeting Ligands to Assist Nanosystems in Locating CAFs

In order to realize the therapeutic effects of nanosystems targeting CAFs, the first step is to locate fibroblasts in the tumor tissue. To achieve this goal, many nanosystems have ligands that specifically bind to CAFs biomarkers [87,90]. FAP-α is a marker specific to CAFs, and the FAP monoclonal antibody (mAb) can specifically bind to it and be used as a targeting ligand. For instance, Lang et al. loaded CXCL12 silencing siRNA (siCXCL12) using a cell-penetrating peptide (CPP) that was adsorbed to an anti-FAP-α monoclonal antibody on the surface. The constructed delivery system effectively reduced the CXCL2 expression in FAP+ CAFs, thereby reducing the migration and metastasis mediated by CAFs [90]. Since most monoclonal antibodies have a high molecular weight, they preclude the formation of nanoparticles with smaller sizes. Compared with monoclonal antibodies, single-chain variable fragments (scFv) have a lower molecular weight and better penetrability, and they are more widely used in nanosystems. Li et al. labeled scFv-Z@FRT and confirmed that it had a high binding capacity with tumor tissue [86]. The authors found that scFv-Z@FRT aggregated around CAFs and led to their clearance and the degradation of ECM in the tumor [86]. In addition to the antigen–antibody reaction, some artificially synthesized peptides can also be used as FAP-targeting ligands. For example, cleavable amphiphilic peptide (CAP) can be specifically recognized and cleaved by FAP-α to release an internal drug that exerts a therapeutic effect. Ji et al. used a CAP that self-assembled into nanoparticles carrying doxorubicin to eradicate CAFs in prostate cancer [12], subsequently promoting tumor cell apoptosis and inhibiting tumor growth in vivo. Tenascin C, a tumor-specific protein mainly secreted by CAFs in most cancers, is another biomarker [91,92]. The short peptide FHKHKSPALPSVGGG (FH) is its ligand, and is widely used as the targeting ligand for CAFs [91]. Chen et al. prepared nano-liposomes modified by FH peptide, which specifically induced the apoptosis of CAFs [77]. Compared with control nano-liposomes without FH peptide modification, those with FH were more absorbed by CAFs and were highly cytotoxic [77]. In summary, the key step in realizing the therapeutic role of nanosystems is the recognition of CAFs, and FAP has been the most commonly used biomarker for this purpose. FAP monoclonal antibodies and small-molecule peptides, such as scFv and CAP, can be used in nanosystems. To identify CAFs, the ligand of Tenascin C, FH peptide, can also be used.

### 4.2. Carrying Systems Promoting Drug Penetration and Absorption in CAFs

Nanoparticles are frequently used in drug-carrying systems. Such nanoparticles deliver drugs to CAFs and then degrade, thereby releasing the drugs to eliminate CAFs or regulate their function to exert therapeutic effects. Currently, commonly used materials mainly include biocompatible polymers, liposomes, and ferritin [87]. Biocompatible polymers usually include cellulose and chitosan, as well as artificially synthesized dendrimers conjugated to drugs, and they degrade to release drugs that play a therapeutic role when entering the CAFs [82,84]. Ernsting et al. prepared the Cellax-DTX polymer, which was conjugated to docetaxel, polyethylene glycol (PEG), and acetylated carboxymethylcellulose, and injected it into a mouse model of pancreatic cancer [75]. They found that the polymer eliminated 90% of SMA+ CAFs and significantly decreased tumor proliferation and metastasis [75]. Further, since liposomes have a high biocompatibility, low immunogenicity, and relatively simple preparation, they have been used in CAF-targeted nanosystems [93]. In combination with specific ligands, liposomes can target CAFs to deliver a variety of drugs, including nucleic acids and short peptides, that regulate the functions of CAFs. For example, Chen et al. prepared 90 nm liposomes by thin lipid film hydration, which effectively entered the tumor tissue to eliminate CAFs and reduced the ECM and IFP in the stroma [94]. Additionally, ferritin is an iron storage protein whose self-assembled subunit structure can form a cavity for encapsulating drugs [95]. Through special surface modification, it can be used to target CAFs to deliver photosensitizers for subsequent photodynamic therapy. For instance, ferritin loaded with the photosensitizer ZnF_16_Pc effectively eliminated CAFs in allogeneic transplanted breast cancer, inhibited the growth and metastasis of A549 tumors in nude mice, and enhanced the anti-tumor immune response [87]. Together, different drug-carrying systems, including biocompatible polymers, liposomes, and ferritin, can promote the entry of drugs, and, when combined with special surface modification, they can improve the action of drugs on CAFs to enhance their therapeutic effects.

### 4.3. Different Cargoes Eliminate or Regulate Functions of CAFs

A variety of drugs or small molecular agents can be carried by nanosystems and enter CAFs to exert therapeutic effects. At present, nanosystems can carry a variety of cargo, including chemotherapeutic drugs, nucleic acids, and short peptides, to directly kill CAFs or regulate their functions [12,92,93]. Cytotoxic drugs are the most commonly carried drugs [81,82]. Insoluble drugs carried by nanosystems can penetrate the barrier formed by CAFs, which can then be eliminated to further promote the entry of drugs and exert anti-tumor effects. For instance, Zhu et al. used glycolipid-based polymeric micelles (GLPMs) to deliver telmisartan and doxorubicin to breast cancer cells, which significantly reduced the α-SMA + CAF population, attenuated the solid stress in the tumor and the tumor vessel pressure, and inhibited tumor growth in vivo [84]. However, chemotherapeutic drugs can also have potential side effects. FAP is expressed not only on the surface of CAFs but also in multipotent bone marrow stem cells, and some therapeutic approaches, such as immunotherapy designed to bind FAP, lead to bone marrow suppression [15,96]. In order to reduce the occurrence of adverse reactions, photodynamic therapy (PDT) can be used for the targeted removal of CAFs. Zinc hexafluorophthalocyanine (ZnF_16_Pc) is one of the most commonly used photosensitizers. It can enter CAFs to generate ^1^O_2_, which has cell-killing effects [97]. In another study, Li et al. injected a nanosystem containing ZnF_16_Pc into tumor-bearing mice and irradiated the tumor with a 671 nm laser, which significantly reduced the positive staining of α-SMA and the synthesis of ECM [86]. 

However, the massive death or depletion of CAFs in the stroma does not always result in the expected therapeutic effect. Interestingly, the extensive depletion of CAFs caused an increase in molecules of the damage response program (DRP), such as Wnt 16, which increased the secretion of inflammatory factors and reduced the effectiveness of chemotherapy [14]. Therefore, another strategy for CAF-oriented nanotherapy could be based on regulating CAF-related functions or reducing their activity. Nucleic acids, peptides, and other small-molecule drugs carried by nanosystems have exhibited potential anti-tumor effects by regulating the function of CAFs. For instance, Gao et al. prepared cyclic RGD (cRGD)-miR-22-sponge nanoparticles to neutralize miR-22 in the exosomes of breast cancer CAFs, and they found that a decrease in miR-22 enhanced the therapeutic effect of tamoxifen [73]. In addition, metal sodium nanoparticles, such as gold nanoparticles (GNPs), can regulate the function of CAFs without relying on drugs, making it a potential method of anti-tumor therapy (Figure 3). For instance, GNPs induced the up-regulated expression of fatty acid synthesis genes in CAFs, including fatty acid synthetase (FASN), sterol regulatory element-binding protein 2 (SREBP2), and fatty acid-binding protein 3 (FABP3) genes, and increased the lipid content in cells, thereby transforming CAFs into cells with a static phenotype with a high fat content and low proliferation [80].

In summary, a variety of nanomaterials, including biocompatible polymers, liposomes, and ferritin, can be used to deliver drugs and facilitate their entry into tumor tissues. The modification of targeting ligands allows them to specifically bind to CAFs and exert an anti-tumor effect (Table 3). In order to reduce the occurrence of adverse reactions or treatment failures, the two main alternative treatment strategies are photodynamic therapy, to locally eliminate CAFs, or the delivery of small-molecule targeted drugs, to regulate the function of CAFs.

## 5. Conclusions

In this review, we summarize the use of CAF-targeted biomaterials and nanomaterials in cancers, from basic research to clinical practice. A variety of materials, including biochips, 3D culture models, and nanosystems, provide powerful tools for the identification of CAF expression profiles, the in vitro simulation of biological behaviors, and targeted therapy research. However, research on these biological materials still needs to overcome many obstacles.

Firstly, CAFs and fibroblasts are heterogeneous [2], where the results obtained from biochips might be influenced by various factors, such as the controls used for the comparison and biochip selection. Therefore, it is necessary to re-verify these rapid results at multiple levels and using multiple methods. At the same time, a variety of normal controls should be used to prevent false-positive results.

Secondly, in current co-cultivation studies, cell lines are often used to construct tumor models, and their cell composition is relatively simple, which makes it difficult to replicate the actual in vivo conditions of tumors [98,99]. Recently, patient-derived tumor organoids (PDOs), a technology that combines decellularization technology, 3D co-cultivation, and microfluidic devices, have emerged. PDOs are believed to retain the inhibitory properties of tumor cells and better simulate the physical and chemical properties of the natural ECM [13], and they are increasingly being used to research different cancers [100,101,102,103]. Whether they will affect the treatment of tumors requires verification in future studies on PDOs.

Thirdly, although nanomaterials or nanosystems have shown promise in killing or inhibiting CAFs and tumor cells, studies have shown that solely removing tumor cells or CAFs may lead to treatment resistance, necessitating new strategies to improve these approaches. As explored in our previous research, the simultaneous targeting or sequential target perturbation of cancer cells and CAFs can increase the anti-tumor effect [2,104]. However, whether this treatment strategy can be applied to nanotherapies remains to be verified.

## Figures and Tables

**Figure 1 ijms-22-11671-f001:**
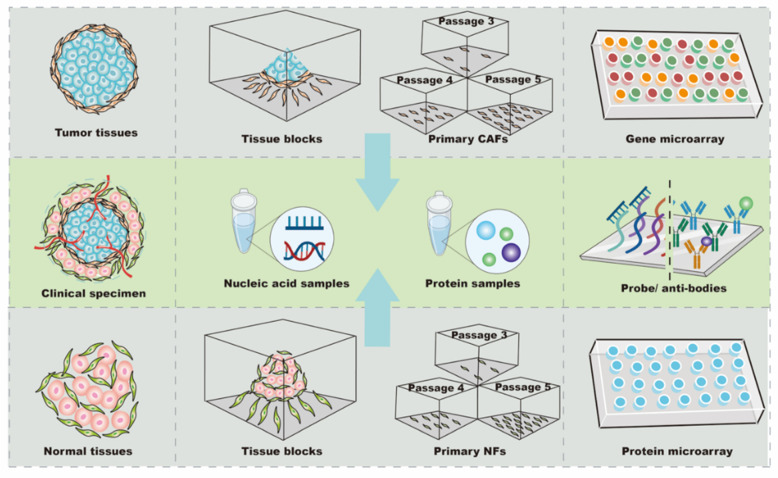
The biochips targeting CAFs in cancers. Tumor tissues and normal tissues are derived from clinical samples. Based on the difference in the adherence time of fibroblasts and tumor cells, CAFs or NFs are isolated and purified from tissue blocks for primary culture. Generally, the 3–5 passages cells are used for subsequent research, and nucleic acid samples or protein samples are extracted by commercial kits or centrifugation for subsequent testing. The large number of probes or antibodies fixed on the surface of the gene chip or protein chips specifically bind to nucleic acids or proteins in the sample to analyze differences in expression profiles.

**Figure 2 ijms-22-11671-f002:**
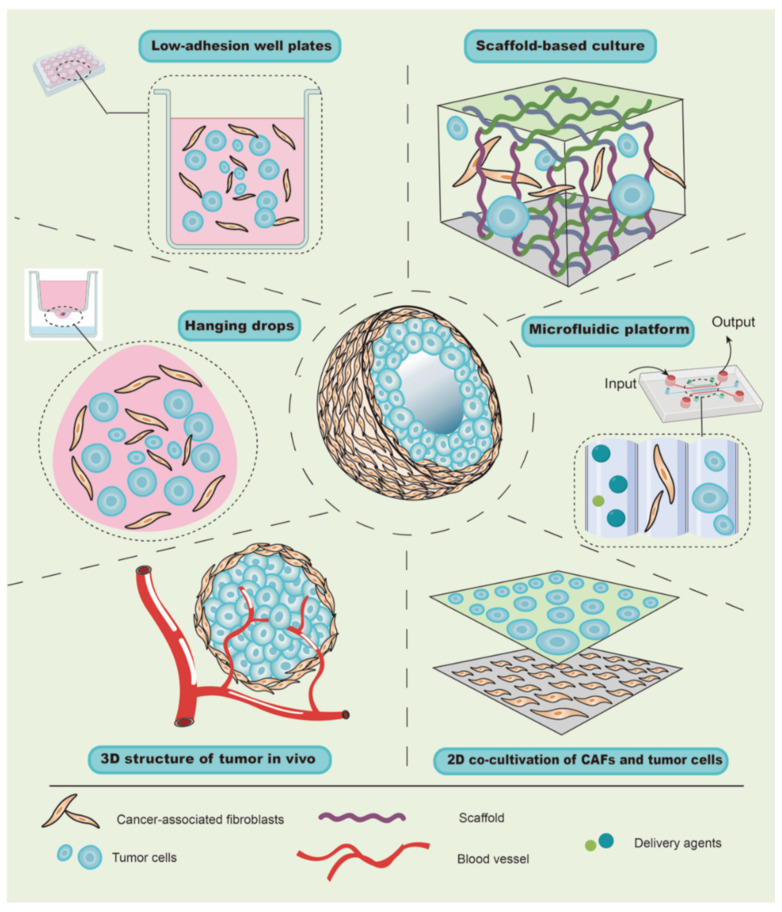
Co-culture models of CAFs and tumor cells. Tumors in vivo have a complex structure, where the tumor cells and CAFs crosstalk with each other in 3D structure. The tumor cells or CAFs in traditional co-cultured model grow in 2D space. The usage of hanging drops aims to form tumor spheroids. Cell suspension flows out from the small holes of the upper container and forms hanging drops under the action of surface tension, and then the cells inside gradually form spheres under the action of the tumor. The liquid in the lower container is performed to compensate for the evaporation of water in the hanging drops. Using the low-adhesion well plates, which make cells unable to adhere to the wall, the suspended cells aggregate into visible spheroids. Various scaffold materials form complex 3D structures for cell culture by providing space. The gel containing tumor cells/CAFs is poured into the channels etched on carrier, and combined with the perfusion system. The cells in the channels can grow under the fluid pressure, thereby simulating the interstitial fluid flow of the tumor.

**Figure 3 ijms-22-11671-f003:**
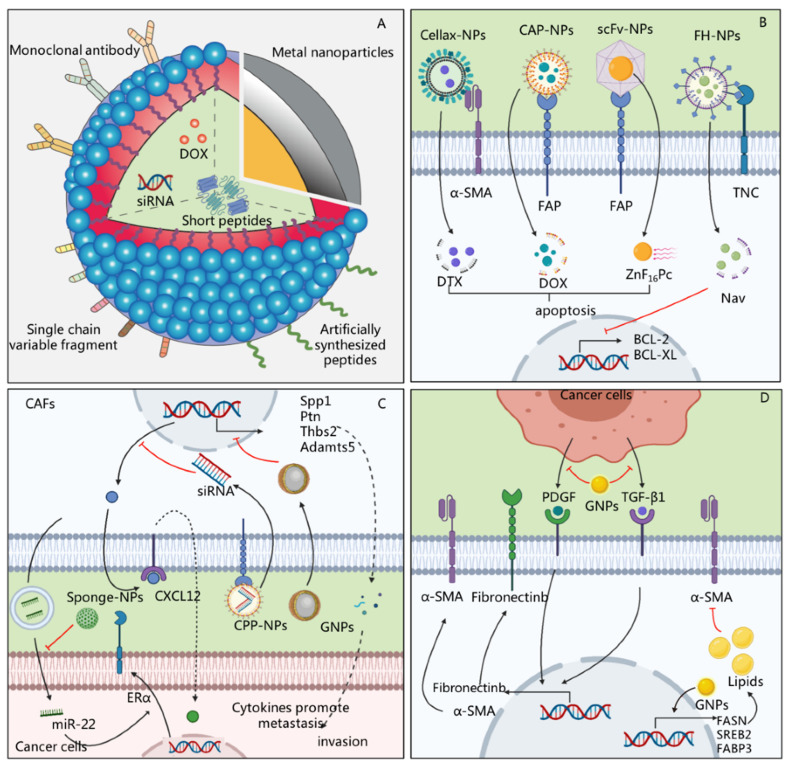
The nanomaterials targeting CAFs. (**A**) Illustration of the basic structure of nanosystems targeting CAFs. The nanosystems mainly consist of three parts: targeting ligands, drug-carrying systems, and cargo. Metal nanoparticles are mainly composed of metal particles with tiny sizes. (**B**) Targeting ligands including cellax, cleavable amphiphilic peptide (CAP), single-chain variable fragments (scFv), and FHKHKSPALPSVGGG (FH) in the nanoparticles specifically bind to α-smooth muscle actin (α-SMA), fibroblast-activated protein (FAP), and tenascin C (TNC) on the surface of CAFs to promote the entry of chemotherapeutic drugs, including doxorubicin (DOX) and docetaxel (DTX), photosensitizers zinc hexafluorophthalocyanine (ZnF_16_Pc), or short peptides navitoclax (Nav) into cells, and target the killing of CAFs or up-regulate the expression of BCL-2 and BCL-XL to increase the apoptosis of CAFs. (**C**) The sponge nanoparticles (sponge-NPs) neutralize miR-22 in the exosomes of CAFs, which inhibit the expression of estrogen receptor-α (ER-α) and reduce therapeutic resistance. The siRNA in CPP-NPs decrease the expression of C–X–C motif chemokine ligand 12 (CXCL12), subsequently reducing the cancer metastasis of tumor. Gold nanoparticles (GNPs) exhibit a suppressive role in tumor invasion, inhibiting expression of osteopontin (Spp1), pleiotrophin (Ptn), thrombospondin-2 (Tnbs2), and ADAM metallopeptidase with thrombospondin type 1 motif 5 (Adamts5). (**D**) GNPs inhibit the expression of α-SMA and fibronectinb in CAFs by inhibiting the platelet-derived growth factor (PDGF) and transforming growth factor-β1 (TGF-β1) expression in the cancer cells. It also induces the up-regulated expression of fatty acid synthesis genes in CAFs, including fatty acid synthetase (FASN), sterol regulatory element-binding protein 2 (SREBP2), and fatty acid-binding protein 3 (FABP3) genes, increases the lipid content, therefore inducing CAFs to stay in a quiescent state, and inhibits tumor-promoting functions. Back arrows: promotion; red “T” arrows: inhibition.

**Table 1 ijms-22-11671-t001:** Biochips targeting cancer-associated fibroblasts in different human cancers.

Tumor	Samples	Controls	Targets	Extraction Method	Platform (Probe)	Array Results (Up/Down-Regulated)	Validation Results (Methods)	Ref.
NSCLC	15	2 cm from primary	mRNA	RNase kit	Affymetrix	22/24	14 mRNAs (RT-PCR)	[27]
COAD	3	Skin and liver	mRNA	N/A	Affymetrix	170/203 (vs. skin); 18/14 (vs. liver)	9 mRNAs (RT-PCR)	[19]
HNSC	3	Skin/buccal mucosa	mRNA	RNA STAT-60	GEArray	1/0	MT1-MMP (WB)	[26]
HNSC	3	2 cm from primary	mRNA	N/A	Agilent	100/0	N/A	[22]
STAD	N/A	5 cm from primary	mRNA	Trizol Reagent	Agilent	10/10	FGF9 (WB, IHC)	[21]
BC	6	CAFs without treatment	mRNA	RNeasy kit	HumanHT 12 v4	35/0	IL-8 (RT-PCR, WB)	[28]
GBC	65	Adjacent normal tissues	mRNA	GeneChip Kit	Affymetrix	466/596	RT-PCR: NOX1 (RT-PCR)	[29]
BC	N/A	CAFs without treatment	miRNA	MirVana kit	TaqMan	7/22	miR-338-3p (RT-PCR)	[10]
BC	N/A	NFs cell line	miRNA	FlashTag Kit.	Affymetrix	1/0	miR-3613-3p (RT-PCR)	[7]
BLCA	5	Bladder/foreskin	miRNA	MirVana kit	miRXplore	0/2 (vs. foreskin); 2/0 (vs. bladder)	5 miRNAs (RT-PCR)	[25]
UCEC	5	Adjacent normal tissues	miRNA	Trizol Reagent	Affymetrix	7/8	5 miRNAs (RT-PCR)	[30]
AH	72	Normal breast tissue.	miRNA	MirVana Kit	Agilent	9/5	miR-200b/c (RT-PCR)	[18]
CHOL	2	Skin	miRNA	MirVana kit	TaqMan	162/93	3 miRNAs (RT-PCR)	[24]
HGSOC	67	Normal ovaries	LncRNA	N/A	Affymetrix	17/22	N/A	[31]
ESCA	49	CAFs without treatment	Protein	Centrifuged	RayBio	5/0	PAI-1 (WB, ELISA)	[17]
CRC	3	Colorectal mucosa	Protein	Filtered	RayBio	34/3	4 proteins (IHC)	[20]

AH: atypical hyperplasia, BC: breast cancer, BLCA: bladder urothelial carcinoma, CHOL: cholangiocarcinoma, COAD: colon adenocarcinoma, CRC: colorectal cancer, ELISA: enzyme-linked immunosorbent assay, ESCA: esophageal carcinoma, FGF9: fibroblast growth factor 9, GBC: gallbladder cancers, HGSOC: high grade serous ovarian cancer, HNSC: head and neck squamous cell carcinoma, IHC: immunohistochemistry, IL-8: interleukin-8, MT1-MMP: membrane type 1-matrix metalloproteinase, N/A: not available, NOX1: nicotinamide adenine dinucleotide phosphate oxidase 1, NSCLC: non-small-cell lung cancer, PAI-1: plasminogen activator inhibitor-1, RT-PCR: reverse transcription polymerase chain reaction, STAD: stomach adenocarcinoma.

**Table 3 ijms-22-11671-t003:** Nanoparticle targeted cancer-associated fibroblasts in anti-cancer therapies.

Nanomaterials	Payload	Tumors	Source of CAFs	Target	Effect on CAFs	Ref
GNPs	N/A	OC	Primary culture	N/A	Inhibit the activation	[78]
GNPs	N/A	OSCC	Primary culture	N/A	Inhibit the migration, activity, and communication	[79]
GNPs	N/A	PDAC	Primary culture	N/A	Transform to quiescence	[80]
CAP-NP	DOX	PC	CAFs cell lines	FAP	Selective apoptosis	[12]
rGO nanosheets	DOX	CC	FAP + CAFs	FAP	Cell-killing	[81]
HA@DSP-pep-DSP	DOX	PC	CAFs cell lines	FAP	Cell-killing	[82]
FH-NB-DOX	DOX	PC	WPMY-1 cells	Tenascin C	Eradication	[83]
GLPM	Tel/DOX	BC	NIH/3T3 cells	α-SMA	Decrease CAF population	[84]
Cellax-DTX polymer	DTX	PDAC	Xenograft	α-SMA	Depletion of CAFs and increase tumor perfusion	[75]
HSA-PTX@CAP-ITSL	HSA-PTX	PDAC	NIH/3T3 cells	FAP	Cell-killing	[85]
Z@FRT	ZnF_16_Pc	BC	Xenograft	FAP	Eradication of CAFs by PDT	[86]
αFAP-Z@FRT	ZnF_16_Pc	BC	Xenograft	FAP	Eradication and stimulates anti-CAFs immunity	[87]
FH-SSL-Nav	Nav	HCC	LX-2 cells lines	Tenascin C	Selective apoptosis	[77]
TR-PTX/HCQ-Lip	PTX and HCQ	PDAC	Integrin αvβ3+ CAFs	Integrin αvβ3+	Inhibit autophagy in CAFs	[88]
LPD	sTRAIL	PDAC	NIH/3T3 cells MRC-5 cells	N/A	Revert CAFs to quiescent state	[89]
PNP/siRNA/mAb nanosystem	siRNA	PC	CAFs cell lines	FAP	Downregulate CXCL12 expression in CAFs	[90]
cRGD-miR-22-sponge nanoparticles	miR-22 sponge	BC	CD63 + CAFs	N/A	Inhibit therapeutic resistance by CD63 + CAF miR-22	[73]

α-SMA: α-smooth muscle actin, BC: breast cancer, CAFs: cancer-associated fibroblasts, CAP: cleavable amphiphilic peptide, CC: colon cancer, cRGD: cyclic RGD, CXCL12: C–X–C motif chemokine ligand 12, DOX: doxorubicin, DTX: docetaxel, DSP: doxorubicin-ss-polyamidoamine, FAP: fibroblast-activated protein, FH: FHKHKSPALSPVGGG, FH-SSL-Nav: Nav-loaded nanoliposomes modified with peptide FH, GLPM: glycolipid-based polymeric micelles, GNPs: gold nanoparticles, HA: hyaluronic acid, HCC: hepatocellular carcinoma, HCQ: hydroxychloroquine, HSA: human serum albumin, Lip: liposomes, LPD: lipid-coated protamine DNA complexes, mAb: monoclonal antibodies, N/A: not available, Nav: navitoclax, NB: nanobubble, NPs: nanoparticles, OSCC: oral squamous cell carcinoma, PC: prostatic cancer, PDAC: pancreatic ductal adenocarcinoma, PDT: photodynamic therapy, PNP: peptide nanoparticles, PTX: paclitaxel, rGO: reduced graphene oxide, Tel: telmisartan, TRAIL: TNF-related apoptosis-inducing ligand, TR: TH-RGD, ZnF_16_Pc: zinc hexafluorophthalocyanine, Z@FRT: ZnF_16_Pc-loaded ferritins.

## Data Availability

Not applicable.

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
