# Peer review of "Fundamental and Clinical Applications of Materials Based on Cancer-Associated Fibroblasts in Cancers"

_ijms, 2021, doi:10.3390/ijms222111671_

Round 1

Reviewer 1 Report

Very nice review. You might want to think of a more appealing title. 

This paper reviews the current knowledge on CAFs and biomaterials used in their research. The paper starts by explaining in a very useful and practical manner the current sources of CAFs, their culture, identification and analysis. In a second part it reviews the possibilities to culture CAFs in 3D setting with or without scaffolds or in microfluids set ups. In the last part it focus in the nanomaterials that are being used to target CAFs and modulate their phenotype, which might in the future give rise to novel treatment strategies. This paper is based in good and recent publications. I believe it would be a very important source of information for anyone aiming to begin to work with CAFs and also for most experienced researchers. 

Author Response

Response:

Thank you so much for your suggestion and comments. For the title, we have changed it to a new one. Hopefully, it would be better than the previous one.

Revision:

Fundamental and Clinical Applications of Materials Based on Cancer-associated Fibroblasts in Cancers

Reviewer 2 Report

The review addresses relevant issues related to the applications of biomaterials and nanomaterials for identifying selective biomarkers of cancer-associated fibroblasts (CAFs) involved in tumorigenesis, tumor growth, invasion and metastasis as well as presents their potential as targeted drug delivery systems for anti-cancer therapy. The manuscript deserves publication in International Journal of Molecular Sciences. However, there are a number of points that should be mentioned in order to improve the presentation of the review.

Major remarks

  1. 1. The sub-section: “3.3. Microfluidic platform co-cultivation technology in CAFs” is presented very briefly, taking into account the amount of literature data available, and should be considerably expanded.
  2. The legends to Figure 2 and Figure 3 are not sufficiently informative and should be completed. Moreover, the full names of the abbreviations used in Figure 3 should be added to the corresponding legend.
  3. Some of the full names of the abbreviations used in Table 3 are missing and should be completed accordingly.

Minor remarks

  1. Lines 129-130 – the statement: “they are cut into small pieces and digested for the primary culture” requires some revision.
  2. Lines 412-413 – the expression: “… which significantly reduced α-SMA+ CAFs” should be replaced by the expression: “… which significantly reduced α-SMA+ CAF population”.
  3. Table 2 – the term: “concentration” should be replaced by the term: “scaffold concentration”.
  4. Table 3 – the expression: “decrease CAFs” should be replaced by the expression: “decrease CAF population”.
